# ScaleAC: Scale Actor-Critic by Replay Ratio

## Abstract

Employing a high replay ratio, defined as the number of updates of an agent's network parameters per environment interaction, has recently become a promising strategy to improve sample efficiency in reinforcement learning (RL). However, most existing efforts to effectively scale a replay ratio stagnate at small values, leaving the potential of scaling a replay ratio to hundreds underexplored. In this paper, we aim to break the bottleneck of replay ratio scaling to achieve sample-efficient RL. We start from the critical pathology that simply increasing the replay ratio leads to severe dormant neurons in the critic network of actor-critic (AC), which fundamentally undermines the learning process. To address this problem, we propose a novel method called ScaleAC, which is built upon advanced AC algorithms (e.g., REDQ, DrQ-v2). First, ScaleAC introduces a periodic soft network parameter reset to reduce dormant neurons when updating the critic at a high frequency. Second, ScaleAC diversifies the replay experience through two kinds of data augmentation to prevent overfitting. Experiments across diverse MuJoCo and DMC tasks demonstrate that ScaleAC successfully achieves effective RL training at high replay ratios of up to 256 in vector-based RL and 8 in visual pixel-based RL, yielding substantial learning acceleration and performance improvement.

## 1 Introduction

When applying reinforcement learning (RL) to real-world applications, a critical limitation of existing RL approaches is their poor sample efficiency, which requires a huge amount of environment interactions to learn satisfactory policies. Therefore, sample-efficient RL algorithms are essential for practice, as it is always desirable to learn with a minimal amount of environment interactions.

One natural idea is to scale the replay ratio, the number of updates of an agent's parameters for each environment interaction (Chen et al., 2021). Recent studies demonstrate that increasing the replay ratio brings substantial performance improvement in well-tuned RL algorithms (D'Oro et al., 2023; Smith et al., 2023). For example, Chen et al. (2021) introduce an in-target random minimization technique called REDQ into soft actor-critic (SAC) to support a replay ratio up to 20 for continuous control. Subsequently, DroQ (Hiraoka et al., 2022) regularizes the critic with Dropout (Srivastava et al., 2014) and Layer Normalization (Xu et al., 2019) to harvest similar benefits to REDQ with the same replay ratio of 20 at a lower computational overhead. Further advances by Nikishin et al. (2022) found that deep RL agents incur a risk of overfitting to earlier experiences, and simply periodic resetting a part of the agent allows SAC to scale at a high replay ratio of 32. More recently, D'Oro et al. (2023) push the replay ratio of SAC to 128 by proposing Scaled-by-Resetting SAC (SR-SAC), which fully resets the network parameters of SAC within a fixed update interval.

Despite the above works successfully training RL at a high replay ratio, they rarely examine what happens inside the agent network when facing the high update frequency, therefore limiting the possibility of scaling RL to a higher replay ratio. In this paper, we aim to break the bottleneck of scaling replay ratio in the sense of both *scaling efficiency* (the same ratio but higher performance) and *scaling ceiling* (the highest ratio that improves performance monotonically). We establish a connection between the replay ratio scaling and the dormant neuron, showing that increasing the replay ratio leads to a large portion of network neurons becoming inactive in the critic to harm learning. To address this problem, we propose ScaleAC, which is built upon advanced AC algorithms such as REDQ and DrQ-v2 (Yarats et al., 2022), to scale the high replay ratio of agents to a new degree (i.e., 256 in vector-based RL and 8 in pixel-based RL) with two key innovations. First, we utilize the periodic plasticity injection technique (Ash & Adams, 2020; D'Oro et al., 2023) to tackle the

severe dormant neuron problem in the critic at high replay ratios. Second, we integrate two kinds of data augmentation techniques into the high-replay-ratio setting to diversify the input state to prevent overfitting. Through the synergy of two key components, ScaleAC successfully addresses the severe dormant neuron issue at high update frequency and, therefore, scales the AC algorithms to the new recorded replay ratios. Experiments in MuJoCo (Todorov et al., 2012) and DMC (Tunyasuvunakool et al., 2020) environments demonstrate that, compared to various baselines, ScaleAC significantly accelerates learning and improves performance with high replay ratios to achieve sample efficiency.

The main contributions of this paper are summarized below:

- We propose ScaleAC, a novel method that integrates periodic network reset and data augmentation into advanced AC algorithms to enable learning at extremely high replay ratios.
- We provide the experimental analysis showing that high replay ratios lead to a severe dormant neuron phenomenon in the critic, preventing thorough use of high-frequency updates.
- We scale the replay ratio of RL agents to a new record of up to 256 for state-based RL and 8 for pixel-based RL, achieving superior sample efficiency compared to strong baselines.

## 2 BACKGROUND

### 2.1 REINFORCEMENT LEARNING

Consider a Markov Decision Process (MDP). At each discrete time step $t$, an agent in the environment observes a state $s_t$, the agent responds by selecting an action $a_t$, and then the environment provides the next reward $r_t$ and state $s_{t+1}$. For convenience, we use the simpler notations of $r$, $s$, $a$, $s'$, and $a'$ to refer to a reward, state, action, next state, and next action, respectively. The objective of an RL agent is to optimize its policy $\pi$ where $a \sim \pi(s)$ to maximize the expected discounted cumulative reward $J(\pi) = \mathbb{E}_\pi[\sum_{t=0}^T \gamma^t r_t]$, where $\gamma \in [0, 1)$ is a discount factor and $T$ is the horizon. The state-action value function $Q^\pi(s, a) = \mathbb{E}_\pi[\sum_{t=0}^T \gamma^t r_t | s_0 = s, a_0 = a]$ gives the expected return starting in $s$, taking an arbitrary action $a$, then following policy $\pi$. In deep RL, policy and value functions are approximated with deep neural networks.

### 2.2 REDQ

Randomized Ensembled Double Q-Learning (REDQ) (Chen et al., 2021) adopts an in-target minimization across a subset $\mathbb{M}$ of $M$ Q-functions, which is randomly sampled from an ensemble of $N$ Q-functions, to derive a lower update target for reducing overestimation. Based on SAC (Haarnoja et al., 2018) with entropy regularization to encourage exploration, the Q target of REDQ is computed as

$$y = r + \gamma(\min_{i \in \mathbb{M}} Q_i(s', \tilde{a}') - \beta \log \pi(s'|\tilde{a}')), \tilde{a}' \sim \pi(\cdot|s'), \tag{1}$$

where $i$ is the index of Q-functions and $\beta$ is the coefficient of the entropy term in SAC. And the policy $\pi_\theta$ is updated with gradient ascent as

$$\nabla_\theta \frac{1}{N} \sum_{i=1}^N (Q_{\phi_i}(s, \tilde{a}_\theta(s)) - \beta \log \pi_\theta(\tilde{a}_\theta(s)|s)), \tilde{a}_\theta(s) \sim \pi_\theta(\cdot|s), \tag{2}$$

where each Q-function is parameterized by $\phi$ and the policy is parameterized by $\theta$.

### 2.3 DRQ-V2

DrQ-v2 (Yarats et al., 2022) is an advanced off-policy AC algorithm for visual continuous control, which uses data augmentation to learn directly from pixels. DrQ-v2 is updated as Deep Deterministic Policy Gradient (DDPG) (Lillicrap et al., 2016), which concurrently learns a Q-function $Q_\phi$ and a deterministic policy $\mu_\theta$ where $a = \mu_\theta(s)$. The Q target is computed as

$$y = r + \gamma(Q_{\bar{\phi}_i}(s', \mu_\theta(s'))), \tag{3}$$

and the policy is updated with gradient ascent as

$$\nabla_\theta Q_\phi(s, \mu_\theta(s)) = \nabla_{\tilde{a}} Q_\phi(s, \tilde{a})|_{\tilde{a}=\mu_\theta(s)} \nabla_\theta \mu_\theta(s). \tag{4}$$

## 2.4 DORMANT NEURON

Sokar et al. (2023) identify the dormant neuron phenomenon in deep RL, where an agent's network suffers from an increasing number of inactive neurons during the training process, thereby affecting network expressivity. The definition of the dormant neuron is given below.

**Definition 1** ($\tau$-Dormant Neuron). Given an input distribution $D$, let $\rho_j^l(x)$ denote the activation of neuron $j$ in layer $l$ under input $x \in D$ and $N_l$ be the number of neurons in layer $l$. The normalized activation score of a neuron $j$ in layer $l$ is defined as follows:

$$d_j^l = \frac{\mathbb{E}_{x \in D}|\rho_j^l(x)|}{\frac{1}{N_l}\sum_{k=1}^{N_l}\mathbb{E}_{x \in D}|\rho_k^l(x)|}. \tag{5}$$

Then neuron $j$ in layer $l$ is defined as $\tau$-dormant if its score $d_j^l \leq \tau$. In this paper, we set $\tau$ at 0.01.

## 2.5 SCALING REPLAY RATIO IN DEEP REINFORCEMENT LEARNING

Moderately increasing the replay ratio for model-free reinforcement learning algorithms has been shown to be a competitive data-efficient baseline for both discrete and continuous control when compared to model-based reinforcement learning methods (Nikishin et al., 2022; D'Oro et al., 2023). For example, REDQ (Chen et al., 2021) uses ensembles with in-target minimization to stabilize SAC training at a high replay ratio of 20, which achieves the same level of sample efficiency when compared to model-based reinforcement learning algorithms such as MBPO (Janner et al., 2019). Later, Nikishin et al. (2022) found that deep RL agents incur a risk of overfitting to earlier experiences, and simply periodic resetting a part of the agent, such as its last few layers, mitigates this primacy bias problem and allows SAC to achieve its superior performance at the high replay ratio of 32. Next, D'Oro et al. (2023) further scale the replay ratio of RL agents up to 128 by proposing Scaled-by-Resetting SAC (SR-SAC) and Scaled-by-Resetting SPR (SR-SPR) algorithms. For instance, SR-SAC completely resets all agent parameters to initial values every $2.56 \times 10^6$ updates.

A parallel stream of work attempts to scale the model size of RL agents through more advanced network architectures to accommodate high replay ratios. For example, BRO (Nauman et al., 2025) scales the SAC critic to about 5 million parameters, using various tricks such as layer normalization and residual connections, to support a high replay ratio of 10. Similarly, SimBa (Lee et al., 2025) scales up network parameters of SAC through network architecture modifications, including an observation normalization layer, a residual feedforward block, and a layer normalization, to achieve a replay ratio of up to 16. In this work, we focus on scaling the replay ratio with the default network architecture and model size from a new perspective of connecting dormant neurons to replay ratios.

## 3 METHOD

In this section, we build ScaleAC upon REDQ (Chen et al., 2021) for vector-based RL and DrQ-v2 (Yarats et al., 2022) for pixel-based RL at high replay ratios. First, in Section 3.1, we reveal that high replay ratios cause severe dormant neurons in the SAC critic. Second, in Section 3.2, we introduce the soft network reset to tackle the dormant neurons. Third, in Section 3.3, we integrate random amplitude scaling to ScaleAC to diversify state vectors. Fourth, in Section 3.4, we build the visual version of ScaleAC on DrQ-v2 with both soft network reset and data augmentation on image pixels.

### 3.1 THE DORMANT NEURONS IN CRITIC AT HIGH REPLAY RATIOS

Here, we conduct an experimental study in MuJoCo and increase the replay ratio of updating the critic as Chen et al. (2021). Specifically, we measure the dormant neuron ratios, which are defined as the proportion of $\tau$-dormant neurons of a neural network, in the SAC critic. The dormant neuron ratios and test episode returns are shown in Figure 1. Clearly, we observe on all tested tasks that increasing the replay ratio results in high dormant neuron rates in the critic network. At the same time, when the replay ratio reaches a high value, such as 64 or 128, the performance of SAC drops significantly on all tasks. This experimental study reveals that high replay ratios lead to high dormant neuron rates in the critic, which undermines the network's representation ability as a larger portion of network neurons becomes inactive and therefore cripples the learning process of RL agents. This

correlation of replay ratios and dormant neurons motivates us to reduce the dormant neurons in the critic to stabilize the training of RL at high replay ratios for desired sample efficiency. Next, we introduce the periodic plasticity injection technique to tackle the severe dormant neuron problem.

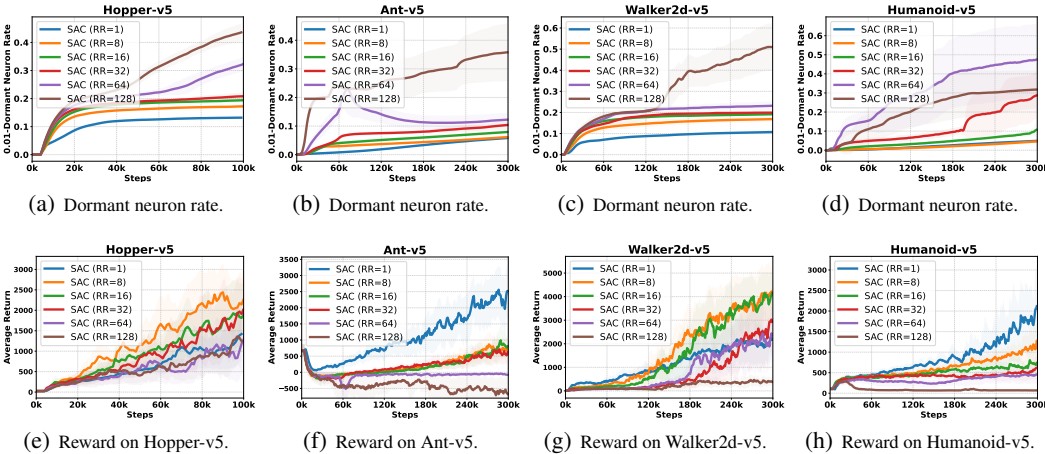

(a) Dormant neuron rate.    (b) Dormant neuron rate.    (c) Dormant neuron rate.    (d) Dormant neuron rate.

(e) Reward on Hopper-v5.    (f) Reward on Ant-v5.    (g) Reward on Walker2d-v5.    (h) Reward on Humanoid-v5.

Figure 1: The dormant neuron rates and test episode returns of SAC with different replay ratios of 1, 8, 16, 32, and 64. The dormant neuron rates increase with the replay ratio on these MuJoCo tasks.

## 3.2 SHRINK & PERTURB TO TACKLE DORMANT NEURONS

Nikishin et al. (2022) choose to fully reset the network parameters of a part of the SAC agent, such as its last few layers, to initial values. Similar strategy is followed by D'Oro et al. (2023) to completely reset all the agent parameters every $2.56 \times 10^6$ of its updates. Such a resetting behavior cleans the learned weights and biases in the reset network layers, and heavily relies on the replay buffer to restore the learned experience of RL agents. Therefore, in ScaleAC, we choose to partially reset the network parameters by interpolating between the current network parameters and the initial network parameters. Specifically, we introduce the Shrink & Perturb strategy (Ash & Adams, 2020) to partially reset the agent network parameters to initial values periodically to maintain the network plasticity. This Shrink & Perturb strategy was originally proposed to warm-start neural network training to incorporate newly arriving data without sacrificing generalization (Ash & Adams, 2020). Recently, Shrink & Perturb has been employed in the SPR (Schwarzer et al., 2021) algorithms, such as SR-SPR (D'Oro et al., 2023) and BBF (Schwarzer et al., 2023), to prevent overfitting under a high replay ratio setting for discrete-action-space control. It has also been applied into the domain of multiagent RL (Yang et al., 2024) and large language model post-training (Liu et al., 2025). Differently, in this work, we focus on integrating Shrink & Perturb into AC algorithms for continuous-action-space control. The formulation of Shrink & Perturb is defined as

$$\theta_t \leftarrow \alpha\theta_t + (1 - \alpha)\theta_0, \tag{6}$$

and

$$\phi_t \leftarrow \alpha\phi_t + (1 - \alpha)\phi_0, \tag{7}$$

where $\theta_0$ is an agent's initial policy network parameters and $\phi_0$ is the initial critic network parameters. $\theta_t$ and $\phi_t$ are the current agent policy and critic network parameters, respectively. The interpolation factor $\alpha$ decides how much the current network parameters are kept.

## 3.3 RANDOM AMPLITUDE SCALING FOR DATA AUGMENTATION IN VECTOR-BASED RL

As ScaleAC updates RL agents with a high replay ratio, it is natural to augment each mini-batch of transitions sampled from the replay buffer to prevent overfitting. Therefore, we introduce the random amplitude scaling (Laskin et al., 2020) into ScaleAC to diversify the replay experiences and prevent overfitting under high replay ratios (Yang et al., 2024; Ma et al., 2024). Random amplitude scaling is a classical data augmentation technique, especially for state-based RL with proprioceptive

inputs (e.g., positions and velocities) (Laskin et al., 2020; He et al., 2023), which randomizes the amplitude of input states while keeping intrinsic consistencies. Its formulation is defined as

$$
\begin{aligned}
s &\leftarrow s * z, \\
s' &\leftarrow s' * z,
\end{aligned}
\tag{8}
$$

where $z \sim U(z_a, z_b)$ is randomly sampled from a uniform distribution over $[z_a, z_b]$. Note that the random amplitude scale is applied randomly across the batch experiences but consistently across time, i.e., the same randomization to the current and next input state vectors. With random amplitude scaling, the intrinsic consistencies are kept, such as the sign of inputs along adjacent time steps.

Now we give the detailed algorithm of ScaleAC for vector-based RL, which is shown in Algorithm 1. Lines 8-9 use the in-target random ensemble minimization from REDQ to calculate the target Q-value. In Line 7, the random amplitude scaling is performed on the sampled mini-batch transitions. The critic is updated $N_{RR}$ times every environment interaction step. As indicated in Line 14, the actor is updated every $T_\theta$ critic updates or after the $N_{RR}$ critic updates. If we set $T_\theta$ at a larger number, such as 20 in MuJoCo, the actor is updated less frequently than the critic. In Lines 18-19, Shrink & Perturb is conducted every $T_R$ environment steps on agent networks, which suffer from the severe dormant neuron problem when updated at high replay ratios.

---

**Algorithm 1** ScaleAC for Scaling Vector-Based Actor-Critic by Replay Ratio

---

1: Initialize policy network parameters $\theta$, the critic network parameters $\phi_i, i = 1, 2, \cdots, N$, and an empty replay buffer $D$. Set target critic network parameters $\bar{\phi}_i \leftarrow \phi_i, i = 1, 2, \cdots, N$. Set agent network reset interval $T_R$. Set policy network update interval $T_\theta$.
2: **for** each time step $t$ **do**
3:     Agent takes action $a_t \sim \pi_\theta(\cdot|s_t)$. Step into state $s_{t+1}$. Receive reward $r_t$.
4:     Add transition data to the replay buffer: $D \leftarrow D \cup \{(s_t, a_t, r_t, s_{t+1})\}$.
5:     **for** each update time $n_{RR}$ from 1 to $N_{RR}$ **do**
6:         Sample a mini-batch $B = \{(s, a, r, s')\}$ from $D$.
7:         Apply random amplitude scaling as in Equation (8) on sampled transition batch $B$.
8:         Sample a set $\mathbb{M}$ of $M$ distinct indices from $\{1, 2, \cdots, N\}$.
9:         Compute the target Q-value $y$ (same for all critics):

$$
y = r + \gamma(\min_{i \in \mathbb{M}} Q_{\bar{\phi}_i}(s', \tilde{a}') - \beta \log \pi_\theta(\tilde{a}'|s')), \tilde{a}' \sim \pi_\theta(\cdot|s').
\tag{9}
$$

10:         **for** $i = 1, 2, \cdots, N$ **do**
11:             Update $\phi_i$ with gradient descent using

$$
\nabla_{\phi_i} \frac{1}{|B|} \sum_{(s,a,r,s') \in B} (Q_{\phi_i}(s, a) - y)^2.
\tag{10}
$$

12:             Update target networks with $\bar{\phi}_i \leftarrow \rho\bar{\phi}_i + (1 - \rho)\phi_i$.
13:         **end for**
14:         **if** ($n_{RR} \mod T_\theta = 0$) or ($n_{RR} = N_{RR}$) **then**
15:             Update policy network parameters $\theta$ with gradient ascent using

$$
\nabla_\theta \frac{1}{|B|} \sum_{s \in B} (\frac{1}{N} \sum_{i=1}^{N} Q_{\phi_i}(s, \tilde{a}_\theta(s)) - \beta \log \pi_\theta(\tilde{a}_\theta(s)|s)), \tilde{a}_\theta(s) \sim \pi_\theta(\cdot|s).
\tag{11}
$$

16:         **end if**
17:     **end for**
18:     **if** $t \mod T_R = 0$ **then**
19:         Perform Shrink & Perturb as in Equation (6) and (7) on agent networks.
20:     **end if**
21: **end for**

---

## 3.4 Extending ScaleAC to Visual Pixel-Based RL

In this section, we extend ScaleAC to an advanced pixel-based AC algorithm, DrQ-v2 (Yarats et al., 2022), to improve the replay ratio to a new degree in visual RL with image input. First, we apply the in-target minimization technique in REDQ to DrQ-v2, where the Q target is computed as in Deep

Deterministic Policy Gradient (DDPG) (Lillicrap et al., 2016) that

$$y = r + \gamma(\min_{i \in \mathbb{M}} Q_{\bar{\phi}_i}(s', \mu_\theta(s'))), \tag{12}$$

where $\mu_\theta$ is a parameterized actor function which deterministically maps states to a specific action, and the agent policy is updated through the chain rule by maximizing $Q$. $\mathbb{M}$ is a subset of $M$ Q-functions, which is randomly sampled from the ensemble of size $N$.

Shrink & Perturb is also employed in the visual version ScaleAC. For the image-based state input, we utilize another data augmentation technique called random shift with bilinear interpolation (Yarats et al., 2022), which applies random shifts to pixel observations for image augmentation. In the visual continuous control of DMC, the random shift is instantiated by first padding each side of $84 \times 84$ observation rendering by 4 pixels (by repeating boundary pixels), and then selecting a random $84 \times 84$ crop, yielding the original image shifted by $\pm 4$ pixels. Then the bilinear interpolation is applied on top of the shifted image by replacing each pixel value with the average of the four nearest pixel values. The algorithm of this visual version ScaleAC based on DrQ-v2 is detailed in Appendix A.

Next, we experiment with ScaleAC to validate it in high-replay-ratio settings on various continuous control tasks in both the MuJoCo and DMC environments. The detailed hyperparameter settings of ScaleAC for each environment, such as how to reset agent networks, are provided in Appendix C.

## 4 EXPERIMENTS

In this section, we experiment with the proposed ScaleAC in both the MuJoCo (Todorov et al., 2012) and DeepMind Control Suite (DMC) (Tunyasuvunakool et al., 2020) environments of continuous action control. First, in Section 4.1, we benchmark ScaleAC in MuJoCo with the standard SAC algorithm and advanced SAC algorithms such as REDQ and SR-SAC, which support high replay ratios. Second, in Section 4.2, we compare all the algorithms with the same replay ratio and show that ScaleAC also achieves the best performance while maintaining the lowest dormant neuron rates. Third, we further benchmark ScaleAC with baselines on the 7 challenging hard tasks in DMC in Section 4.3. Fourth, as shown in Section 4.4, we show that ScaleAC is able to scale the replay ratio to even 256. Fifth, we give the ablation study in Section 4.5 to validate each component of ScaleAC. Finally, we extend ScaleAC to the domain of visual RL to improve the replay ratio in Section 4.6.

### 4.1 BENCHMARK EXPERIMENTS IN MUJOCO

First, we benchmark ScaleAC and baselines in the classical MuJoCo tasks, including Hopper-v5, Ant-v5, Walker2d-v5, and Humanoid-v5, with details of each task in Appendix B. For baselines, SAC has a standard replay ratio of 1. REDQ is set to its default replay ratio of 20. SR-SAC's replay ratio is 128 according to its recommended configurations. For ScaleAC, we found that a replay ratio of 64 consistently performs well. For Shrink & Perturb, we set $T_R = 2000$ to reset the critic network every 2000 environment steps and set $\alpha = 0.8$ to mix 80% of the values of current network parameters and 20% of the values of initial network parameters. For random amplitude scaling, we set $z_a = 0.8$ and $z_b = 1.2$ for the amplitude range. The policy network update interval $T_\theta$ is 20. The benchmark results in MuJoCo are shown in Figure 2. The reported metrics are averaged over 6 independent trials with different random seeds, and the 95% confidence intervals are shadowed.

As we can see, ScaleAC, with a replay ratio of 64, consistently outperforms baselines within the same number of environment interactions, showing its superior sample efficiency. At the same time, SR-SAC with a higher replay ratio than ScaleAC does not perform well in these MuJoCo tasks, indicating that how to reset RL agents to support a high update frequency is not trivial.

### 4.2 BENCHMARKING WITH THE SAME REPLAY RATIO TO EVALUATE SCALING EFFICIENCY

Next, we also evaluate each method with the same replay ratio of 32 to compare their scaling efficiency. Results of the test episode return and dormant neuron rate are plotted in Figure 3.

As shown in Figure 3, under the same replay ratio of 32, ScaleAC also achieves the best performance among the four MuJoCo tasks, while REDQ is competitive to ScaleAC in Walker2d-v5. When checking the dormant neuron rates in the critic, ScaleAC has the lowest dormant neuron rates,

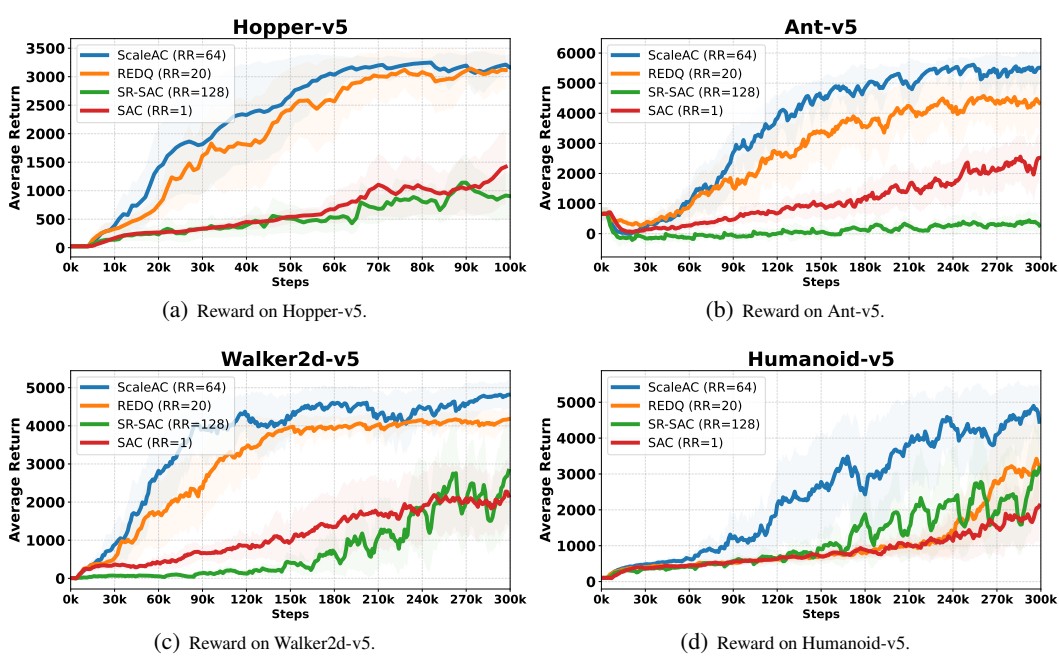

Figure 2: Benchmark RL algorithms with their default replay ratios in the MuJoCo environment. SAC has a replay ratio of 1. REDQ has a replay ratio of 20. SR-SAC has a replay ratio of 128.

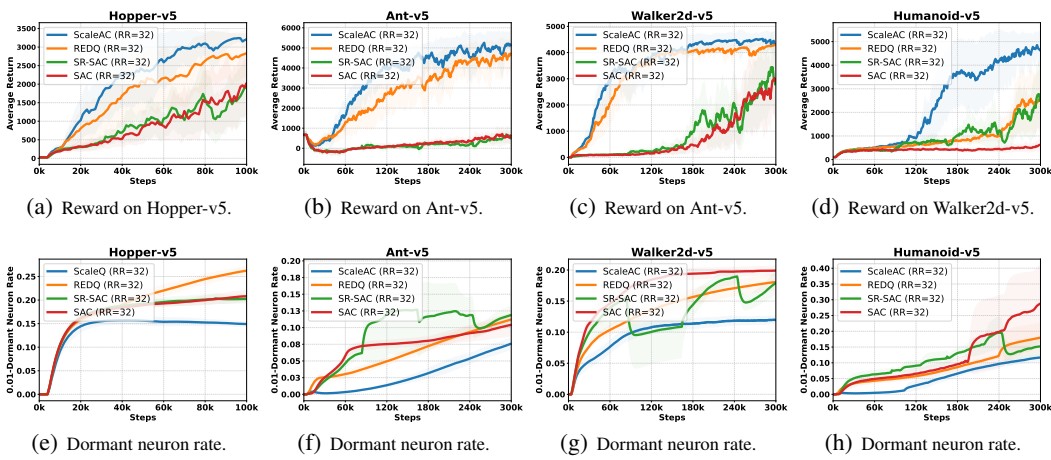

Figure 3: Experimental results of RL methods in MuJoCo with the same replay ratio of 32.

allowing the network to maintain plasticity and learn progressively in the high-replay-ratio setting. On the other hand, fully resetting both the policy network and critic network in SR-SAC does not work well, as it cannot steadily reduce the dormant neurons, especially in the Ant-v5 task. This indicates that introducing the Shrink & Perturb strategy in ScaleAC is the key to stabilizing RL training by maintaining a low level of dormant neuron rate when facing a high update frequency.

### 4.3 BENCHMARK EXPERIMENTS IN DMC

In this section, we validate ScaleAC in DMC, which features a range of locomotion and manipulation tasks. We benchmark methods in 7 challenging DMC Hard tasks, including dog-run, dog-trot, dog-stand, dog-walk, humanoid-run, humanoid-stand, and humanoid-walk. More details about these tasks can be found in Appendix B. Similar to the benchmark experiments in MuJoCo, SAC's replay

ratio is 1, REDQ's replay ratio is 20, and SR-SAC has a replay ratio of 128. For the dog-series tasks, we found that ScaleAC with a replay ratio of 64 performs best. For the humanoid-series tasks, we found that a replay ratio of 256 works best for ScaleAC. The specific hyperparameters of ScaleAC in DMC are given in Appendix C, where $T_R = 2000$, $\alpha = 0.8$, $z_a = 0.8$, and $z_b = 1.2$ are the same as in MuJoCo. The results of each method with its corresponding replay ratio are demonstrated in Figure 4. The reported test episode returns are averaged over 6 independent trials with different random seeds, and the 95% confidence intervals are shadowed. We also provide additional experiments on three simple walker-series tasks in DMC, which are available in Appendix D.

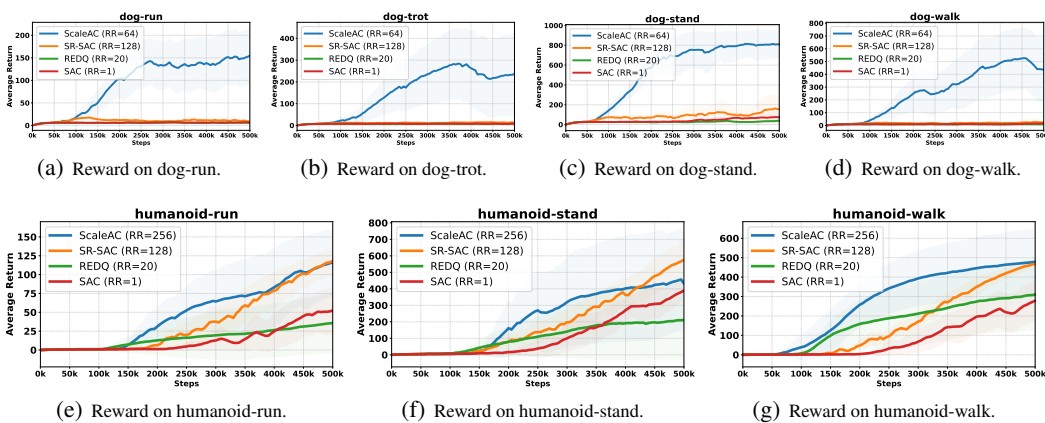

(a) Reward on dog-run. (b) Reward on dog-trot. (c) Reward on dog-stand. (d) Reward on dog-walk.

(e) Reward on humanoid-run. (f) Reward on humanoid-stand. (g) Reward on humanoid-walk.

Figure 4: Benchmark RL algorithms with their default replay ratios in the DMC environment. SAC has a replay ratio of 1. REDQ has a replay ratio of 20. SR-SAC has a replay ratio of 128.

In Figure 4, ScaleAC outperforms baselines in almost all DMC Hard tasks, with one exception that ScaleAC performs slightly worse than SR-SAC on humanoid-stand in the last environment steps. Meanwhile, REDQ struggles to learn these DMC Hard tasks, which necessitates ScaleAC's components beyond REDQ, including Shrink & Perturb and random amplitude scaling. While SR-SAC achieves good performance on humanoid-series tasks, it fails on dog-series tasks, showing that fully resetting networks may not be a general solution to train RL at high replay ratios. The impressive performance of ScaleAC in both MuJoCo and DMC environments demonstrates its effectiveness in stabilizing RL training under high replay ratios to accelerate learning. At the same time, ScaleAC consistently achieves superior performance in all the tested tasks, demonstrating its broad generality.

### 4.4 Scaling up the Replay Ratio to 256 to Evaluate Scaling Ceiling

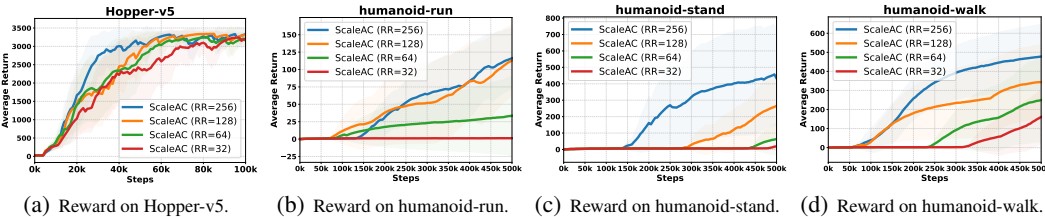

(a) Reward on Hopper-v5. (b) Reward on humanoid-run. (c) Reward on humanoid-stand. (d) Reward on humanoid-walk.

Figure 5: Scaling up the replay ratio of ScaleAC to 256 on Hopper-v5 in the MuJoCo environment and on humanoid-run, humanoid-stand, and humanoid-walk in the DMC environment.

In this section, we demonstrate that ScaleAC is able to scale up the replay ratio to even 256 to speed up the learning process. The results of ScaleAC with different replay ratios of 32, 64, 128, and 256 are plotted in Figure 5. As we see, ScaleAC significantly accelerates the learning of RL agents with a replay ratio of up to hundreds, especially on humanoid-stand and humanoid-walk. To the best of our knowledge, this is the highest reported replay ratio to successfully train deep RL in the current literature (Ma et al., 2025). In summary, ScaleAC unlocks the potential to achieve better sample

efficiency through scaling of the replay ratio to hundreds (i.e., 256). On the other hand, we should also notice that a higher replay ratio does not mean higher performance. ScaleAC may have different optimal replay ratios on different tasks. For example, in the dog-series tasks, the best replay ratio of ScaleAC is 64, while replay ratios such as 128 and 256 decrease the performance of ScaleAC. The replay ratio scaling of ScaleAC in the dog-series tasks is shown in Appendix E.

### 4.5 ABLATION STUDY OF SCALEAC

Here, we conduct the ablation study to verify each component in ScaleAC. The ablation results are given in Figure 6. We see that both Shrink & Perturb and random amplitude scaling contribute to the performance of ScaleAC. Especially, without Shrink & Perturb, ScaleAC fails to learn in dog-run and dog-stand, which necessitates Shrink & Perturb to reduce dormant neurons when the replay ratio is high. Meanwhile, random amplitude scaling also enhances the performance of ScaleAC.

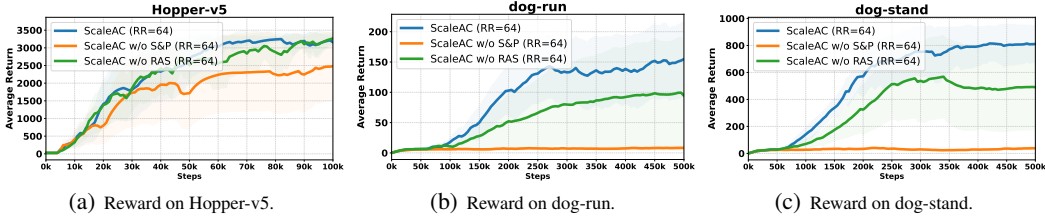

(a) Reward on Hopper-v5.    (b) Reward on dog-run.    (c) Reward on dog-stand.

Figure 6: The ablation study of ScaleAC. 'ScaleAC w/o S&P' indicates removing the Shrink & Perturb in ScaleAC. 'ScaleAC w/o RAS' indicates removing random amplitude scaling in ScaleAC.

### 4.6 BENCHMARK EXPERIMENTS OF VISUAL SCALEAC IN VISUAL DMC

In this section, we compare the visual ScaleAC algorithm in visual DMC (details in Appendix B.3) with two advanced visual RL approaches, DrQ-v2 (Yarats et al., 2022) with a default replay ratio of 0.5 and Adaptive RR (Ma et al., 2024) with a replay ratio increasing from 0.5 to 2. Results are shown in Figure 7. The visual ScaleAC achieves substantial learning acceleration and performance improvement with a higher replay ratio. Notably, in hopper-hop, the visual ScaleAC boosts learning with a replay ratio of 8, demonstrating the great potential of high replay ratios in visual RL.

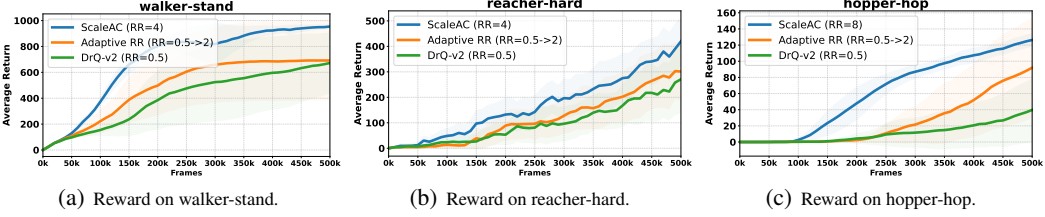

(a) Reward on walker-stand.    (b) Reward on reacher-hard.    (c) Reward on hopper-hop.

Figure 7: Benchmarking results on visual DMC tasks.

## 5 CONCLUSION

In this paper, we propose ScaleAC to scale up the replay ratio to hundreds (i.e., 256) in RL. We found that high replay ratios lead to saturated dormant neurons in the critic, thus undermining RL learning. To tackle this problem, we integrate Shrink & Perturb into advanced AC algorithms such as REDQ and DrQ-v2 to partially reset the critic periodically. Two kinds of data augmentation are also applied to enhance state input diversity. Extensive experiments in MuJoCo and DMC show that ScaleAC greatly improves sample efficiency in both vector and pixel-based RL at high replay ratios.

For future work, first, dynamically adjusting the replay ratio is promising to reduce the computation cost. Second, introducing ScaleAC to LLM post-training also has great potential. Third, introducing plastic network structures to ScaleAC is a natural direction to further improve the replay ratio.

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

# A  SCALEAC BUILT ON DRQ-V2 FOR PIXEL-BASED RL

In this section, we give the detailed algorithm of ScaleAC based on DrQ-v2 for pixel-based RL, which is shown in Algorithm 2. Lines 8-9 use the in-target random ensemble minimization to calculate the target Q-value as REDQ. In Line 7, the random shift with bilinear interpolation is performed on the sampled mini-batch transitions. The critic is updated $N_{RR}$ times every environment interaction. As indicated in Line 14, the actor is updated every $T_\theta$ critic updates or after the $N_{RR}$ critic updates. In Lines 18-19, Shrink & Perturb is conducted every $T_R$ environment steps on agent networks, which suffer from the severe dormant neuron problem with a high replay ratio.

---

**Algorithm 2** ScaleAC for Scaling Pixel-Based AC by Replay Ratio

---

1: Initialize policy network parameters $\theta$, the critic network parameters $\phi_i, i = 1, 2, \cdots, N$, and an empty replay buffer $D$. Set target critic network parameters $\bar{\phi}_i \leftarrow \phi_i, i = 1, 2, \cdots, N$. Set agent network reset interval $T_R$. Set policy network update interval $T_\theta$.
2: **for** each time step $t$ **do**
3:     Agent takes action $a_t \sim \pi_\theta(\cdot|s_t)$. Step into state $s_{t+1}$. Receive reward $r_t$.
4:     Add transition data to the replay buffer: $D \leftarrow D \cup \{(s_t, a_t, r_t, s_{t+1})\}$.
5:     **for** each update time $n_{RR}$ from 1 to $N_{RR}$ **do**
6:         Sample a mini-batch $B = \{(s, a, r, s')\}$ from $D$.
7:         Apply random shift with bilinear interpolation on sampled transition batch $B$.
8:         Sample a set $\mathbb{M}$ of $M$ distinct indices from $\{1, 2, \cdots, N\}$.
9:         Compute the target Q-value $y$ (same for all critics):

$$y = r + \gamma(\min_{i \in \mathbb{M}} Q_{\bar{\phi}_i}(s', \mu_\theta(s'))). \tag{13}$$

10:         **for** $i = 1, 2, \cdots, N$ **do**
11:             Update $\phi_i$ with gradient descent using

$$\nabla_{\phi_i} \frac{1}{|B|} \sum_{(s,a,r,s') \in B} (Q_{\phi_i}(s,a) - y)^2. \tag{14}$$

12:             Update target networks with $\bar{\phi}_i \leftarrow \rho\bar{\phi}_i + (1-\rho)\phi_i$.
13:         **end for**
14:         **if** ($n_{RR} \bmod T_\theta = 0$) or ($n_{RR} = N_{RR}$) **then**
15:             Update policy network parameters $\theta$ with gradient ascent using

$$\frac{1}{|B|} \sum_{s \in B} (\frac{1}{N} \sum_{i=1}^{N} \nabla_{\tilde{a}} Q_{\phi_i}(s, \tilde{a})|_{\tilde{a}=\mu_\theta(s)} \nabla_\theta \mu_\theta(s)). \tag{15}$$

16:         **end if**
17:     **end for**
18:     **if** $t \bmod T_R = 0$ **then**
19:         Perform Shrink & Perturb as in Equation (6) and (7) on agent networks.
20:     **end if**
21: **end for**

---

For the visual version of ScaleAC, it has an encoder to encode the image into a vector for both the actor and critic networks. We also reset the encoder in the same way as resetting the critic.

# B  ENVIRONMENT DETAILS

## B.1  MUJOCO

We consider a total of 4 continuous control tasks for the MuJoCo benchmark. These tasks include Hopper-v5, Ant-v5, Walker2d-v5, and Humanoid-v5. The short descriptions, observation dimension, and action space dimension are listed in Table 1. For ScaleAC and baselines in MuJoCo, we utilize the official codebase of REDQ (Chen et al., 2021) with PyTorch to implement algorithms.

Table 1: Descriptions of Different MuJoCo Tasks.

| Task | Robot | Short Description | State dim | Action dim |
|------|-------|------------------|-----------|------------|
| Hopper-v5 | 2D Runners | 2D monoped for hopping | 11 | 3 |
| Ant-v5 | Quadruped | 3D quadruped for running | 105 | 8 |
| Walker2d-v5 | 2D Runners | 2D biped for walking | 17 | 6 |
| Humanoid-v5 | Humanoid Bipeds | 3D humanoid for running | 348 | 17 |

## B.2 DMC

We consider a total of 7 continuous control tasks for the DMC Hard benchmark (Tunyasuvunakool et al., 2020). These tasks include dog-run, dog-trot, dog-stand, dog-walk, humanoid-run, humanoid-stand, and humanoid-walk. The observation dimension and action space dimension are listed in Table 2. For ScaleAC and baselines in DMC, we utilize the official codebase of SR-SAC (D'Oro et al., 2023) with JAX to implement these algorithms. We additionally list three walker-series tasks in Table 2, which are used in Appendix D for extra benchmarking experiments in DMC.

Table 2: Observation and Action Dimensions for Different DMC Hard Tasks.

| Task | Difficulty | Description | State dim | Action dim |
|------|-----------|-------------|-----------|------------|
| Dog-run | Hard | A Pharaoh Dog model to run | 223 | 38 |
| Dog-trot | Hard | A Pharaoh Dog model to trot | 223 | 38 |
| Dog-stand | Hard | A Pharaoh Dog model to stand | 223 | 38 |
| Dog-walk | Hard | A Pharaoh Dog model to walk | 223 | 38 |
| Humanoid-run | Hard | A 21-joint humanoid to run at 10 m/s | 67 | 24 |
| Humanoid-stand | Hard | A 21-joint humanoid to stand at 0 m/s | 67 | 24 |
| Humanoid-walk | Hard | A 21-joint humanoid to walk at 1 m/s | 67 | 24 |
| Walker-run | Medium | An improved planar walker to run | 24 | 6 |
| Walker-stand | Easy | An improved planar walker to stand | 24 | 6 |
| Walker-walk | Easy | An improved planar walker to walk | 24 | 6 |

## B.3 VISUAL DMC

We consider three tasks in the visual DMC, including walker-stand, reacher-hard, and hopper-hop, to validate the visual ScaleAC. Tasks are summarized in Table 3. In this setting, environment observations are stacks of 3 consecutive RGB images of size $84 \times 84$, stacked along the channel dimension to enable inference of dynamic information like velocity and acceleration. For ScaleAC and baselines in visual DMC, we utilize the official codebase of DrQ-v2 (Yarats et al., 2022) and Adaptive RR Ma et al. (2024) with PyTorch to implement these algorithms.

Table 3: Descriptions of Different Visual DMC Tasks.

| Task | Traits | Difficulty | Action dim |
|------|--------|-----------|------------|
| Walker-stand | stand, dense | easy | 6 |
| Reacher-hard | reach, dense | medium | 2 |
| Hopper-hop | move, dense | medium | 4 |

## C HYPERPARAMETERS OF SCALEAC

In this section, we provide the hyperparameters of ScaleAC. For the Shrink & Perturb strategy, we set $T_R = 2000$ to reset the agent networks every 2000 environment steps and set $\alpha = 0.8$ to keep 80% of the values of current network parameters. For random amplitude scaling, we set $z_a = 0.8$ and $z_b = 1.2$ for the amplitude range. Specifically, in MuJoCo, we set the policy network update interval $T_\theta$ at 20, which is updated less frequently than the critic network. Therefore, we only reset the critic network in MuJoCo. Meanwhile, in DMC, we set the policy network update interval $T_\theta$ at 1 and reset both the policy network and critic network. In visual DMC, we set the policy network

update interval $T_\theta$ at 1 and reset the encoder network, policy network, and critic network. The setting of the random shift with bilinear interpolation for visual ScaleAC is the same as in DrQ-v2. Other hyperparameters are kept the same as the original configurations provided in the official codebases (Chen et al., 2021; D'Oro et al., 2023; Yarats et al., 2022; Ma et al., 2024). The study of hyperparameter sensitivity of ScaleAC is provided in Appendix F for reference.

## D   BENCHMARK EXPERIMENTS ON DMC WALKER-SERIES TASKS

We also conduct additional benchmark experiments in DMC to validate the generality of ScaleAC. We experiment on three walker-series tasks, including walker-run, walker-stand, and walker-walk, with 0.2 million environment steps. The results of each method on these tasks are plotted in Figure 8. All the methods achieve good performance. Meanwhile, although the final performance of SR-SAC is close to ScaleAC, it is clear that ScaleAC learns much faster at the early stage than SR-SAC in these walker-series tasks. Notably, ScaleAC with a replay ratio of 32 also learns faster than SR-SAC with a replay ratio of 128, indicating the superior sample efficiency of ScaleAC.

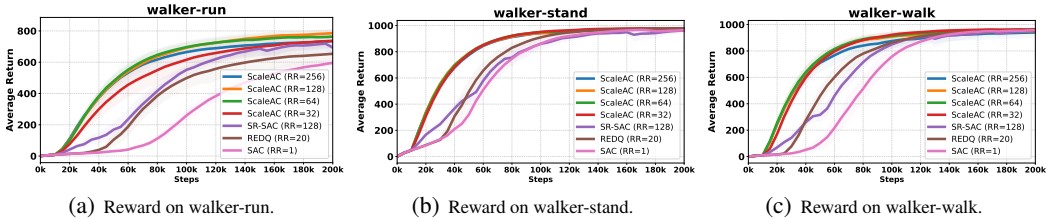

(a) Reward on walker-run.        (b) Reward on walker-stand.        (c) Reward on walker-walk.

Figure 8: Additional benchmarking results on walker-series tasks in DMC.

## E   REPLAY RATIO SCALING OF SCALEAC IN DOG-SERIES TASKS

In this section, we show that a higher replay ratio does not always correspond to higher performance. The replay ratio scaling of ScaleAC on dog-series tasks is plotted in Figure 9. It is clear that 64 is the optimal replay ratio on these tasks. Higher values, such as 128 and 256, hurt the performance. Therefore, we infer that there exists a saturation point of the replay ratio that exhausts a model's plasticity when fitting the given replay buffer. Shrink & Perturb tries to recover the model's plasticity, while random amplitude scaling reduces the plasticity cost of each trained sample, which coincides somewhat with the findings in the domain of visual reinforcement learning (Ma et al., 2024).

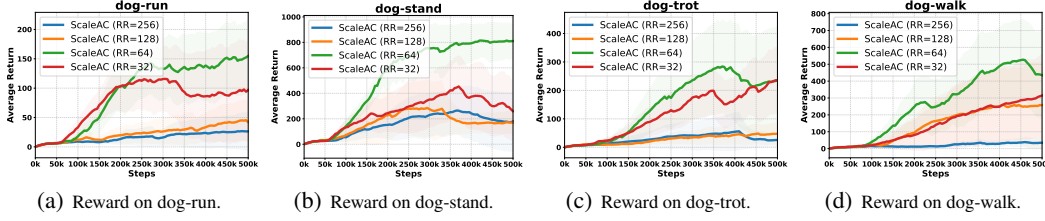

(a) Reward on dog-run.        (b) Reward on dog-stand.        (c) Reward on dog-trot.        (d) Reward on dog-walk.

Figure 9: Replay ratio scaling of ScaleAC on dog-run, dog-stand, dog-trot, and dog-walk in DMC.

## F   HYPERPARAMETER SENSITIVITY

In this section, we study the hyperparameter sensitivity specifically in ScaleAC. First, we show how the interpolation factor $\alpha$, which determines how much of the current network parameters is mixed with the initial network parameters, affects the performance of ScaleAC in Appendix F.1. Second, in Appendix F.2, we investigate the sensitivity of the reset interval $T_R$. Finally, we also show how the amplitude range $[z_a, z_b]$ in random amplitude scaling affects ScaleAC in Appendix F.3. The default values in ScaleAC include $\alpha = 0.8$, $T_R = 2000$, and $[z_a = 0.8, z_b = 1.2]$.

## F.1 THE INTERPOLATION FACTOR IN SHRINK & PERTURB

We study the interpolation factor $\alpha$ with different values, and the corresponding results are given in Figure 10. When $\alpha = 0.0$, the network parameters are fully reset to initial values. When $\alpha = 1.0$, the network parameters are totally kept. As shown in Figure 10, $\alpha = 0.8$ achieves the best performance in the three scenarios tested, which is also the default value across tasks and environments. In the dog-run and dog-stand tasks, $\alpha$ with other values does not perform well, indicating that the interpolation factor $\alpha$ is an important hyperparameter to tune in ScaleAC carefully.

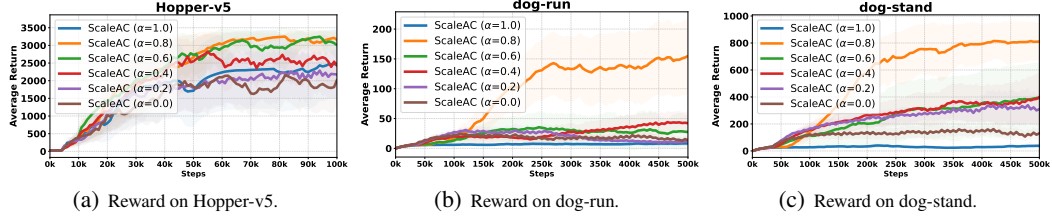

(a) Reward on Hopper-v5.   (b) Reward on dog-run.   (c) Reward on dog-stand.

Figure 10: The hyperparameter sensitivity study on the interpolation factor $\alpha$ of ScaleAC.

## F.2 THE RESET INTERVAL IN SHRINK & PERTURB

The default reset interval $T_R$ is 2000, which means ScaleAC applies Shrink & Perturb every 2000 environment steps. Here we also experiment with $T_R = 500, 1000, 4000$, and 8000. The resulting plots are given in Figure 11. When $T_R = 4000$, ScaleAC achieves the best performance in Hopper-v5 and dog-run, but performs worse than $T_R = 2000$ in the task of dog-stand. Generally, the default reset interval $T_R = 2000$ performs well in these tested cases.

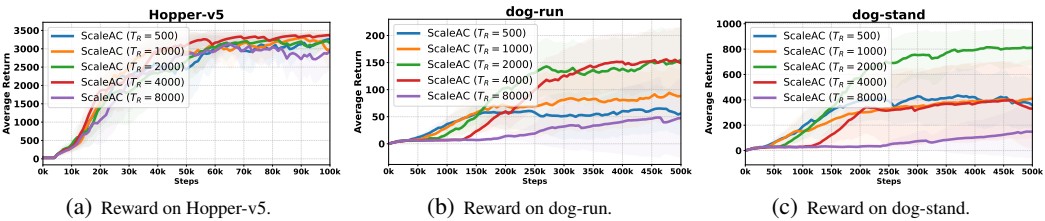

(a) Reward on Hopper-v5.   (b) Reward on dog-run.   (c) Reward on dog-stand.

Figure 11: The hyperparameter sensitivity study on the reset interval $T_R$ of ScaleAC.

## F.3 THE SCALING RANGE IN RANDOM AMPLITUDE SCALING

We study the amplitude scaling range $[z_a, z_b]$ with different values, and the corresponding results are given in Figure 12. We see that, the best scaling range changes in different tasks. For example, in Hopper-v5, $[0.0, 2.0]$ achieves the highest average return while performing sub-optimally in dog-stand. At the same time, the default setting of $[0.8, 1.2]$ consistently performs well in three tasks.

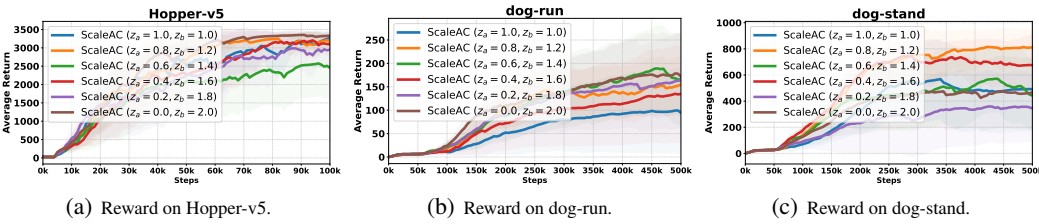

(a) Reward on Hopper-v5.   (b) Reward on dog-run.   (c) Reward on dog-stand.

Figure 12: The hyperparameter sensitivity study on the amplitude range $[z_a, z_b]$ of ScaleAC.

