# OpenReview forum: "ScaleAC: Scale Actor-Critic by Replay Ratio"
_ICLR.cc/2026/Conference — ICLR 2026 Conference Withdrawn Submission_

### Official Review · Reviewer_CB3D · 2025-10-19

**Soundness:** 3
**Presentation:** 3
**Contribution:** 2
**Rating:** 4
**Confidence:** 4

**Summary:**

This paper identifies that high replay ratios in actor-critic methods cause a "dormant neuron" phenomenon, which hinders the learning process. To address this, the authors propose ScaleAC, an algorithm that enables stable training at high update frequencies. ScaleAC integrates two key components: a periodic soft network parameter reset to maintain network plasticity, and data augmentation to prevent overfitting. This combination allows the agent to train effectively at record-high replay ratios of up to 256, resulting in learning acceleration and improved sample efficiency on continuous control benchmarks.

**Strengths:**

1. The method proposed by the authors is simple yet effective, substantially improving training performance under high replay ratios through periodic soft network parameter resets and data augmentation, effectively addressing the core issue of the paper.
2. Experiments are relatively comprehensive, demonstrating the effectiveness of the method in both state-based and pixel-based tasks. The benchmarks tested align with common practices in the field, and the ablation study also showcases the contributions of the two components, with clear logic.
3. The paper is well-structured, with clear language and well-designed figures that effectively convey the research content and results.

**Weaknesses:**

1. **Missing discussion of key related work**: Previous studies have explored similar issues, such as the paper "DrM: Mastering Visual Reinforcement Learning through Dormant Ratio Minimization", which also builds on DrQ-v2 and employs techniques like shrink & perturb to reduce neuron dormancy in pixel-based RL. Additionally, the paper "Neuroplastic Expansion in Deep Reinforcement Learning" discusses enhancing network plasticity and reducing neuron dormancy using Neuroplastic Expansion. The authors should discuss these related works and clearly delineate how ScaleAC differs from and improves upon these methods, particularly DrM.
2. **Complex and Task-Specific Hyperparameter Configuration**: A potential weakness of the paper is the task-specific hyperparameter settings. While the authors demonstrate the effectiveness of ScaleAC across benchmarks, the experimental setup relies on different configurations for different environments. For instance, in the visual-based tasks, a replay ratio (rr) of 4 is used for `walker-stand` and `reacher-hard`, whereas an rr of 8 is used for `hopper-hop`. Furthermore, the policy network update interval $T_\theta$ differs between the experiments on the DeepMind Control Suite and MuJoCo Gym. Different rrs are also applied to the `Dog` and `Humanoid` environments. Although the authors provide some ablation studies to justify these individual choices, the paper lacks a general guiding principle for hyperparameter selection. This reliance on ad-hoc tuning could potentially hinder the algorithm's generalizability and practical application to new tasks.

**Questions:**

1. How is ScaleAC on pixel-based tasks different from DrM? Can the authors provide a direct comparison in terms of performance and methodology?
2. Can the authors elaborate on the general principles guiding the selection of hyperparameters such as replay ratio and policy update intervals across different tasks? Is there a way to standardize these settings to enhance the algorithm's applicability to new environments?
3. How does pixel-based ScaleAC perform compared to DrQ-v2 at the same replay ratios? Currently, the paper only compares ScaleAC with rr=4/8 against DrQ-v2 with rr=2 or adaptive rr, which may not provide a fair comparison. Can you include results for DrQ-v2 at rr=4/8 or ScaleAC at rr=2 to better illustrate the advantages of ScaleAC?
4. Can the authors provide results and analysis on time efficiency and computational resource usage for ScaleAC?

---

### Official Review · Reviewer_hFkU · 2025-10-31

**Soundness:** 3
**Presentation:** 3
**Contribution:** 2
**Rating:** 2
**Confidence:** 4

**Summary:**

This paper proposes ScaleAC, a method to scale the replay ratio (updates per environment interaction) in off-policy actor–critic methods. The authors diagnose that simply increasing replay ratio induces a high fraction of dormant neurons in the critic and degrades learning. ScaleAC combines two main changes: 1) periodic soft network reset and 2) data augmentations. Experiments show improved results in state-based and pixel-based benchmarks (Mujoco and DMC) compared to various baselines.

**Strengths:**

- Diagnoses the problem of dormant neuron rates with higher replay ratios and proposes Shrink & Perturb to address it
- Good empirical results in Mujoco and DMC compared to standard baselines like SAC, REDQ
- Ablations are convincing to show that both Shrink & Perturb and data augmentation are important

**Weaknesses:**

- In Figure 2, can you show lines showing the baselines with the same replay ratio e.g. “REDQ (RR=64)”? I know that Figure 3 uses the same replay ratio for everything but it would be helpful to see how REDQ does with RR=64 since ScaleAC uses 64 in Figure 2
- The data augmentations are not novel and fairly standard in most state-of-the-art RL algorithms already (e.g., random shift is fairly common). Are these not used in the baselines by default?
- My main concern is that given that the data augmentations are very standard, the main contribution is resetting network parameters which has been proposed in previous work already and therefore the novelty is very limited

**Questions:**

See weaknesses

---

### Official Review · Reviewer_kwBb · 2025-11-03

**Soundness:** 2
**Presentation:** 3
**Contribution:** 2
**Rating:** 2
**Confidence:** 5

**Summary:**

This paper mainly focuses on training the actor-critic network with high replay ratio, and suggests that training the network with high-replay ratio promotes high dormant neuron ratio which is the main cause of the performance degradation. Especially, the authors pointed out that the number of dormant neurons in critic network gradually increases as training progresses. To resolve this problem, the authors adopt data augmentation and shrink & perturb, and also do not scale up the network size. In the experiment results, the proposed method outperforms SR-SAC, and also shows remarkable performance on dog tasks in Deepmind Control environment.

**Strengths:**

1. The proposed method shows remarkable performance on hard tasks such as dog tasks in Deepmind Control environment. Furthermore, the overall method is quite simple and easy to implement on other actor-critic algorithms.

**Weaknesses:**

1. I think the overall claim is not novel, and most of the problems that this paper proposes are already well-known in this community. In my opinion, I guess the main contribution on remarkable performance comes from the REDQ algorithm, not applying shrink & perturb or data augmentation. Since many existing algorithms already adopted both approaches and also shows the effectiveness, the novelty is quite limited.

2. Why did you apply high RR only to critic? How about the policy network? Furthermore, it would be better to show the number of dormant neurons of the policy network as well. I think the overall explanation on the algorithm is not enough to support each component.

3. I think it would be better to show the decrease of the number of dormant neurons after applying the proposed method, and also show the dormant neurons of REDQ without applying shrink & perturb and random amplitude scaling.

**Questions:**

Already mentioned in the Weaknesses section.

---

### Official Review · Reviewer_McvW · 2025-11-04

**Soundness:** 2
**Presentation:** 2
**Contribution:** 1
**Rating:** 2
**Confidence:** 4

**Summary:**

This paper proposes ScaleAC to scale the replay ratio up to 256 in continuous control RL. The authors identify that high replay ratios cause severe dormant neurons in the critic network, and address this by integrating Shrink & Perturb (periodic soft network reset) and data augmentation into REDQ/DrQ-v2. Experiments on MuJoCo and DMC demonstrate performance improvements at high replay ratios compared to SAC, REDQ, and SR-SAC.

**Strengths:**

- The empirical analysis connecting high replay ratios to dormant neurons (Figure 1) provides useful insights, showing that dormant neuron rates increase with replay ratio while performance degrades at very high ratios (64, 128).
- The method achieves high replay ratios (up to 256 for vector-based RL and 8 for pixel-based RL) with consistent improvements across diverse tasks, demonstrating practical effectiveness in accelerating learning.
- The paper includes comprehensive experiments across multiple benchmarks (MuJoCo, DMC, visual DMC) with detailed ablation studies and hyperparameter sensitivity analysis, showing the contribution of each component.

**Weaknesses:**

1. The baseline comparisons are fundamentally misaligned with the problem being solved. REDQ addresses overestimation bias through ensemble methods, while ScaleAC tackles plasticity loss. The fair comparison should be REDQ+ScaleAC vs. REDQ+other plasticity methods (e.g., ReDo, ReGraMa, Plasticity Injection), not REDQ alone. This makes it impossible to assess whether the gains come from addressing plasticity or simply from combining orthogonal techniques.
2. **Critical missing baselines**: Recent work (SimBa, BRO) has shown that network architecture modifications are key to supporting high replay ratios and maintaining plasticity. The paper acknowledges these works (Section 2.5) but completely fails to compare against them. Without experiments showing ScaleAC+SimBa vs. SimBa alone, or ScaleAC+default-architecture vs. architecture-improved baselines, the claimed contributions cannot be properly evaluated. This is the most serious weakness.

**Questions:**

1. Why not compare "REDQ + ScaleAC" against "REDQ + ReDo/ReGraMa/Plasticity Injection"? Since REDQ solves overestimation while your method addresses plasticity loss, shouldn't you demonstrate that your plasticity solution is superior to existing ones when combined with the same base algorithm?
2. Can you provide experiments comparing ScaleAC built on SimBa/BRO architectures? Since these architectures specifically address plasticity for high replay ratios, demonstrating whether ScaleAC provides additional benefits would validate its necessity and establish whether the approach is complementary or redundant to architectural solutions.

---

### Author Response · Authors · 2025-12-04
**Thanks the reviewers**

Dear reviewers,

Thank you sincerely for your detailed comments and valuable feedback!

We will continue to improve our work based on your suggestions!

Best,

The authors

---

### Note · Authors · 2026-01-06

I have read and agree with the venue's withdrawal policy on behalf of myself and my co-authors.